# Assessing the Costs of Screening for Ovarian Cancer in the United States: An Evolving Analysis

**DOI:** 10.3390/diagnostics10020067

**Published:** 2020-01-25

**Authors:** Justin W. Gorski, McKell Quattrone, John R. van Nagell, Edward J. Pavlik

**Affiliations:** 1Division of Gynecologic Oncology, Department of Obstetrics & Gynecology, University of Kentucky Chandler Medical Center, 800 Rose Street, Lexington, KY 40536-0263, USA; jrvann2@uky.edu (J.R.v.N.); Edward.Pavlik@uky.edu (E.J.P.); 2University of Kentucky College of Medicine, 800 Rose Street, Lexington, KY 40536-0298, USA; McKell.Oliverio@uky.edu

**Keywords:** ovarian cancer, screening, transvaginal ultrasound, cost analysis

## Abstract

The primary objective of this study is to provide an updated analysis of the cost of screening for ovarian cancer in the United States. Here, we use updated information from the University of Kentucky Ovarian Cancer Screening Trial in conjunction with new modifying factors such as U.S. national estimates of the cost of care (Truven Health MarketScan Database), recently published estimates of earnings lost due to ovarian cancer death and estimates of federal income taxes paid on those earnings. In total, 326,998 screens were performed during the Kentucky trial from 1987 to 2019. At a cost of $56 per screen, we estimate that the total base cost to operate the program over the last 32 years is $18,311,888. When accounting for the surgical cost of 381 false-positive cases, the total cost of the screening program increases by $3,030,474. However, these costs are offset by the benefit of treating more early-stage ovarian cancer in the screened population, with a total cost advantage of $4,016,475 at our institution (Kentucky) or $1,525,050 ($725,700–$3,312,650) (U.S.) nationally. Additionally, program costs are offset by approximately $3,549,000 due to the potential earnings gained by the 26 women whose lives have been saved with screening. Furthermore, the cost of the program is offset by the federal tax dollars paid on the recovered earnings and amounts to $383,292. Ultimately, the net adjusted total cost of the Kentucky screening program is an estimated $13,393,595 at our institution or $15,885,020 ($13,978,068–$16,799,083) nationally. Thus, the adjusted cost per screen is an estimated $40.96 in Kentucky or $48.58 ($42.75–$51.37) nationally.

## 1. Introduction

Ovarian cancer is the fifth most common cause of cancer death in women in the United States. In 2019, it is estimated that 22,530 U.S. women will be diagnosed with ovarian cancer, with 13,980 deaths due to this disease [1]. This high fatality rate can be mostly attributed to the high prevalence of advanced stage-disease (70%) at initial diagnosis [2]. Five-year survival rates vary widely between advanced-stage disease (20–30%) and early-stage disease (70–90%), where tumor growth is confined to the ovary [3]. 

Over the last three decades, a concerted effort has been made to develop an accurate, reliable and cost-effective screening modality to detect early-stage ovarian cancer. Four major trials have evaluated the utility of using transvaginal ultrasound to detect ovarian cancer including the Kentucky trial [4], the Prostate, Lung, Colorectal and Ovarian (PLCO) Cancer Screening Randomized Controlled Trial [5], the UK Collaborative Trial of Ovarian Cancer Screening (UKCTOCS) [6] and the Shizuoka Cohort Study of Ovarian Cancer Screening (SCSOCS) [7] trial. Three of these trials (Kentucky, UKCTOCS and SCSOCS) report a shift to early-stage detection, while both the Kentucky and UKCTOCS report a survival advantage for incident cases due to screening [8,9].

The cost of screening question must be considered in terms of price per screen, savings on the cost of clinical care for management of more early-stage ovarian cancer as well as the savings from lost earnings, and federal tax revenue on those earnings, due to lives saved. Previous cancer cost analysis in 2008 identified ovarian cancer as the second most expensive cancer subtype to treat in women [10]. It was estimated that the average first year cost to treat ovarian cancer was $51,548, which was second only to brain cancer, which is estimated to cost $69,908 during the first year. The high cost of treatment of ovarian cancer makes it likely that a shift toward early-stage detection would drive down costs, and the present effort addresses this possibility.

The primary objective of this study was to perform an analysis of the cost of screening for ovarian cancer using updated information from the University of Kentucky Screening Trial, new U.S. estimates of the cost of care for the management of ovarian cancer and recovery of lost earnings due to ovarian cancer. We have previously published a cost analysis of the Kentucky ovarian cancer screening program, which demonstrated a direct cost of $40,731 per life-year gained [11]. Here, we evolve this analysis to account for subsequent screening accruals, newly available information about the national cost of care for disease management and introduce a consideration for lost earnings and federal tax revenue on those earnings.

## 2. Materials and Methods

### 2.1. Identification of Subjects

Women that enrolled in the University of Kentucky Ovarian Cancer Screening Trial from January 1987 to December 2019 were evaluated. The University of Kentucky institutional review board for human studies approved this prospective cohort trial (88-GYN-I3 Transvaginal Ultrasonography As a Screening Method for Ovarian Cancer, IRB #45030, renewed 1 May 2019). Eligibility criteria include: (1) all asymptomatic women ≥50 years old and (2) asymptomatic women ≥25 years old with a documented family history of ovarian cancer in at least one primary or secondary relative. Women with a known ovarian tumor or a personal medical history of ovarian cancer were excluded from enrollment. Informed consent was obtained, and each patient underwent screening with transvaginal ultrasound.

All ultrasound images were reviewed by a board certified gynecologic oncologist. Follow-up of normal and abnormal scans were performed as already published. Follow-up data on all enrolled patients was obtained through coordinated effort with the University of Kentucky Tumor Registry, the Kentucky Cancer Registry and the Kentucky State Department of Vital Statistics.

True-positive screens were considered those with histology-confirmed primary epithelial ovarian cancer, low malignant potential ovarian tumors, non-epithelial or metastatic cancer to the ovary. However, only subjects with invasive primary epithelial ovarian cancer were included in the cost analysis. False-positive screens included benign ovarian histology including serous cystadenoma, endometrioma, mucinous cystadenoma, cystic teratoma, fibroma, thecoma, Brenner tumor, leiomyoma, hydrosalpinx, paratubal cyst or hemorrhagic cyst.

Unscreened women with clinically detected epithelial ovarian cancer who were referred to the University of Kentucky Markey Cancer Center (UKMCC) for treatment between January 1995 and December 2019 were identified using UKMCC Division of Gynecologic Oncology records. This subset of patients served as the control for this study. A starting point of January 1995 was chosen because after this date all patients with a diagnosis of invasive epithelial ovarian cancer at this institution, underwent maximal cytoreduction at the time of surgical staging, received combination platinum/taxane chemotherapy and were treated by gynecologic oncologists using the same treatment algorithms.

### 2.2. Cost Analysis

Cost analysis for this study was performed by amending a step-wise approach [12,13] and has been previously published by the present authors [11]. This new amended approach accounts for the expense of the screening modality, surgical cost of false-positive cases as well as the offsets of treating more early-stage cancers, recovered earnings due to lives saved and federal tax revenue salvaged by screening (Figure 1).

### 2.3. Screening and False-Positive Costs

The total cost of the screens performed throughout the study from January 1987 to December 2019 was calculated by multiplying the cost of each ultrasound by the total number of screens performed. Next, the surgical cost of false-positive cases was calculated. This was accomplished by multiplying the number of false-positive cases detected during the trial by the UK HealthCare estimated cost to perform each diagnostic surgical procedure. These costs were combined to determine the total cost of screens and false-positive cases.

### 2.4. Stage-Specific Disease Treatment Costs

The number of screen-detected ovarian cancer cases was extracted from the University of Kentucky Ovarian Cancer Screening Trial database. For the matched control group, the total number of clinically detected ovarian cancer patients referred to the University of Kentucky Markey Cancer Center (UKMCC) for treatment from 1995 to 2019 were analyzed for stage at initial diagnosis. The number of expected recurrences for the screen-detected and clinically detected groups were determined based on a 15% early-stage recurrence rate and an 80% late-stage recurrence rate [14].

Two different sources for estimating treatment cost were utilized: (1) UK HealthCare rates (July 2019) which reflect costs in Kentucky and (2) Truven Health MarketScan Commercial Claims and Encounters database (2009–2012) as published in Bercow et al. which reflect national costs in the United States [15]. Both of these cost estimates were then applied to the populations of interest to run the analysis [16].

Primary treatment costs for the first year after diagnosis included the cost of cytoreductive surgery and adjuvant or neo-adjuvant carboplatin/paclitaxel chemotherapy. It was assumed that 95% of patients with stage I–II ovarian cancer and 100% with stage III–IV ovarian cancer received chemotherapy.

Recurrence chemotherapy treatment costs were calculated for two additional years of treatment. UK HealthCare recurrence treatment estimates included the cost of carboplatin/paclitaxel and bevacizumab for six additional cycles as well as an additional 12 months of bevacizumab maintenance therapy. Truven Health recurrence treatment costs were calculated by combining the cost of inpatient services from 6 months to 1-year post-op plus the first-year cost of outpatient services and outpatient drugs. This yearly cost was then multiplied by two to determine the cost of two years of recurrence treatment.

The cost advantage of screening was determined by subtracting the total treatment cost of all clinically detected patients from the total treatment cost of screen-detected patients.

### 2.5. Life-Years Gained Due to Screening

The number of patient lives saved due to screening was determined by finding the difference between the 10-year survival of screen-detected and clinically detected cases. The number of life-years gained due to screening was calculated by subtracting the average age of diagnosis of true-positive trial participants from the average female life expectancy in the United States. In order to calculate the total life-years gained as a result of screening, the number of patient lives saved was multiplied by the life-years gained due to screening.

### 2.6. Adjusted Screening Cost per Life-Year Gained

The cost of the study per life-year gained was calculated by dividing the sum of the cost of screens, surgical cost of false positives and cost advantage of treating more early-stage disease by the total number of life-years gained by screening. 

### 2.7. Recovered Earnings and Tax Dollars

The estimate of lost earnings per cancer death was calculated using figures from a recently published analysis of national and state estimates of lost earnings from cancer deaths in the United States [17] where the total lost earnings due to ovarian cancer was divided by the number of ovarian cancer deaths. The resultant value of lost earnings per cancer death is equivalent to the earnings gained per life saved in this analysis. Next, the number of lives saved due to the Kentucky screening program was multiplied by the earnings gained per life saved and taxed at the average federal income tax rate [18].

### 2.8. Adjusted Net Total Screening Cost

The adjusted net total screening cost was calculated by adding the total surgical cost of false-positive cases to the total cost of screening and then subtracting the total cost advantage of treating more early-stage disease, total earnings gained due to screening and the total tax dollars paid by women saved.

### 2.9. Adjusted Cost per Screen

The adjusted net total screening cost was divided by the total number of screens performed during the study to determine the adjusted cost per screen.

## 3. Results

### 3.1. Survival of Screen-Detected versus Clinically Detected Ovarian Cancer Patients

The effectiveness of ultrasound-based ovarian cancer screening is shown by the significantly better disease-specific survival of the screened population enrolled in the Kentucky trial as compared to the unscreened population treated at the University of Kentucky Markey Cancer Center (UKMCC) during that same time frame. Women with screen-detected ovarian cancer had high survival rates of 86 ± 4%, 68 ± 7%, and 65 ± 7% at 5, 10, and 20 years, respectively. These survival rates are significantly different from unscreened women who were diagnosed with clinically detected ovarian cancer at the UKMCC. Clinically detected women at UKMCC have much lower survival rates of 45 ± 2%, 34 ± 2%, and 19 ± 3% at 5, 10, and 20 years, respectively. Using the log-rank test the *p* < 0.0001 (Figure 2).

### 3.2. Cost of Screening and False-Positive Surgeries

The total cost of screening was updated for this analysis from our previous analysis [11] using similar factors that contribute to cost. In total, 28,580 additional screens were performed from June 2017 to December 2019 and are included in this analysis. This increases the total number of screens performed during the trial from 298,418 to 326,998. In total, 128 true-positive cases were detected during the trial. True-positive results include screen-detected primary epithelial ovarian cancer (*n* = 127), low malignant potential tumors (*n* = 26, excluded from survival analysis) and non-epithelial or metastatic cancer to the ovary (*n* = 23). The positive predictive value (PPV) of the transvaginal ultrasound screening modality is 25% and so it is estimated that 381 false-positive cases resulted using the most recent screening algorithm. From UK HealthCare rates, we have determined that the total surgical cost of false-positive results **is** approximately $3,030,474. This increases the total cost of screening from $18,311,888 to $21,342,362 (Table 1).

### 3.3. Treatment Cost Advantage of Screening

In total, 76 screen-detected primary epithelial ovarian cancer subjects were identified. In total, 48 of these patients were diagnosed with early-stage disease and 28 were initially diagnosed with late-stage disease. For purposes of this study, matched extrapolation of the overall stage of diagnosis of the clinically detected control group at the University of Kentucky Markey Cancer Center (UKMCC) resulted in 22 clinically detected early-stage subjects and 54 clinically detected late-stage subjects.

Cost to treat estimates using either Kentucky rates (Table 2A) or those from the national database (Table 2B) demonstrated a distinct treatment cost advantage for the screen-detected group. The primary treatment cost per case is estimated at $33,464 based off of the Kentucky rate and $93,632 ($62,319–$140,140) per case when extracted from the national database. Using the Kentucky financial figures, the estimated total cost of treating all screen-detected cases of ovarian cancer was $7,363,034 compared to $11,379,509 for clinically detected cases (Table 2A). This resulted in a total cost advantage of screening of $4,016,475. Similarly, when the national treatment cost estimates were applied, the total cost advantage was averaged at $1,525,050 with an interquartile range (IQR) of $725,700 to $3,312,650 (Table 2B).

### 3.4. Life-Years Gained Due to Screening and Associated Cost per Life-Year

Clinical outcome measures of the Kentucky screening trial remained the same from our 2018 publication [11]. The 10 year survival of screen-detected cases was 68% but only 34% for clinically detected cases (Figure 2). Thus, we estimate that screening is responsible for saving 26 lives 10 years after diagnosis. Since the mean age of women diagnosed with screen-detected ovarian cancer in the study is 65 years old and the mean life expectancy of women in the United States is 81 years old, we can assume that each saved woman would live an additional 16 years due to screening. Therefore, the overall total life-years gained due to screening is 416 (Table 3).

The updated and adjusted total cost per life-year gained was determined to be $41,649 using the Kentucky estimates and $47,638 ($43,341–$49,560) using the national database values (Table 4 row D). These values take into account the cost of screens ($44,019/life-year), surgical cost of false positives ($7285/life-year) and cost advantage of treating more early-stage disease ($9655 or $3666/life-year).

### 3.5. Lost Earnings and Federal Tax Dollars Recovered

Figures from a recent analysis of national and state lost earnings from cancer deaths in the United States were used to calculate the lost earning per death due to ovarian cancer [17]. The total lost earnings of ovarian cancer patients in 2015 ($1,629,000,000 ($1,582,000,000–$1,679,000,000)) was divided by the total number of ovarian cancer deaths (11,938) during that year to yield the lost earning/death estimate. Thus, the lost earning/death is estimated at $136,500 ($132,518–$140,643) for this analysis (Table 5 row A). 

When we multiply the lost earnings per death by the total number of patient lives saved due to screening, we estimate that $3,549,000 in earnings have been recovered due to our screening program (Table 5 row C). Assuming an average federal income tax rate on earnings gained of 10.8%, we calculate that the total federal income tax recovered due to screening is $383,292 (Table 5 row E).

### 3.6. Adjusted Net Total Screening Program Cost

Next, taking into account the total screening cost, surgical costs of false-positive cases, the cost advantage of treating earlier detected cases as a result of screening, lost earnings gained and the estimate of tax dollars recovered due to screening, we have determined that the updated net cost of the screening program is $13,393,595 ($13,508,308–$13,274,243) (Kentucky) or $15,885,020 ($13,978,068–$16,799,083) (U.S.) Table 6 row F.

### 3.7. Adjusted Cost per Screen

The base cost per screen in the Kentucky Screening program is $56. However, when the surgical cost of false-positive cases is taken into account, the cost rises to $65.27 (Table 7). Furthermore, when the cost advantage of treating earlier detected cases, earnings gained due to lives saved and the recovered federal tax dollars due to lives saved is taken into account, the adjusted net cost per screen falls to $40.96 (Kentucky) or $48.58 (U.S.) (Table 7).

## 4. Discussion

We show here that the base ultrasound screening cost of $56 can be raised and lowered by a variety of modifying factors. When we expand cost considerations to include the expense of false-positive cases, we raise the cost by ~$9. However, when we include the cost advantage of treating more early-stage disease, we decrease the unit cost by ~$13 using UK HealthCare rates. Expansion of our analysis to include national estimates of health care expenses using the Truven MarketScan database results in only a mean unit cost decrease of ~$4 per ultrasound. This makes sense since the national estimates includes places with much higher costs of living than Kentucky. Additionally, when we factor in the national estimates of lost earnings per cancer death from ovarian cancer and the federal income tax collected on those earnings, the unit cost of ovarian screening is further reduced. Consideration of all of these factors combined results in a net overall savings per screen. 

The largest single expense in the Kentucky trial is the cost of the screening ultrasound itself at a total of more than $18,000,000. In the United States, the cost of a transvaginal ultrasound (TVUS) is widely variable and is largely influenced by regional health care cost variation and reimbursement rates negotiated between providers and third-party payers. For example, retail prices of a TVUS in Lexington, Kentucky range from $153 to $1333 but in New York, New York range from $148 to $656 [19]. Importantly, retail pricing is different from the dollars collected for TVUS. In fiscal year (FY) 2018 Medicare collections on TVUS averaged $202 and increased to $206 in 2019 in the UK HealthCare system, while collections from commercial health insurance averaged $272 in FY 2018 and $291 in FY 2019. Thus, the actual reimbursed rate from third-party payers is lower than posted retail rates. In this context, our estimate of $56 per ultrasound screen is much lower because it is a research rate that reflects only direct costs while reimbursements are adjusted for facility costs to operate. It must be kept in mind that the cost of a TVUS exam is much lower than the costs of other radiologic imaging modalities such as computed tomography (CT) and magnetic resonance imaging (MRI) or biochemical assays that need to be performed in a Clinical Laboratory Improvement Amendments (CLIA)-certified laboratories. Overall, TVUS presents no radiation risk, is minimally invasive, reliable, provides information that can be followed over time and is inexpensive when compared to other radiological modalities.

In addition to the unit cost of the TVUS screening, we have considered the surgical cost of false-positive cases and report that this expense is in excess of $3,000,000. We do acknowledge that a false-positive screen may inflict undue psychological distress upon the screen-positive but disease-negative subjects. We have not tried to factor in the cost of this related distress in our analysis. However, we found that over half of the women that we evaluated were characterized by either a no-stress trajectory (30%) or a decreasing-stress trajectory (25%) over four months of follow-up [20]. As an additional consideration, we take the position that the false-positive cases receive the benefit of prophylaxis from the surgical removal of the abnormality. Removal of a large benign cyst, leiomyoma or endometrioma before it becomes symptomatic, may keep some screen-positive but disease-negative women out of emergency rooms. In extreme cases, controlled removal of the abnormal but non-malignant adnexa, may have prevented emergent surgery in the future. Thus, false positives in the screening program may provide risk-reduction that lowers psychological stress.

The most striking financial offset included in this analysis is the benefit of treating more early-stage disease. The difference in stage at initial diagnosis has a significant economic benefit. First, according to National Comprehensive Care Network (NCCN) guidelines, patients with stage IA or IB disease with certain ovarian cancer histopathologies such as low-grade serous or grades 1 and 2 endometrioid ovarian cancer, do not always need adjuvant chemotherapy. [21] Although these histologic subtypes are rare, they do make up approximately 5% of the primary ovarian cancers detected with our TVUS screening program. Thus, treatment for these patients is considered complete after surgical staging. Second, recurrence rates are significantly lower for more common histopathologic subtypes, such as high serous ovarian cancer, when detected at earlier stages. Thus, we have calculated and included a significant cost advantage of detecting more early-stage cancer with the TVUS screening program. This saving is quite sizable when we use the UK HealthCare rates and is in excess of $4,000,000. The national database rates yield a more modest but still sizable cost advantage nationally that is greater than $1,500,000. Additionally, when the range of the national data is considered, this benefit may be in excess of $3,300,000 in higher cost markets. Third, our analysis only included the cost of recurrence chemotherapy for two additional years. However, in many instances, recurrent ovarian cancer patients are candidates for a plethora of cytotoxic agents in an attempt to palliate recurrences over the course of many years. Overall, the financial and survival benefit from detecting early-stage disease should not be underestimated. 

Quantification of the economic impact of potential earnings and federal tax dollars recovered due to screening is challenging. We have included several relevant factors and kept assumptions to a minimum. We included the recovery of federal tax dollars because the earnings gained from screening would at minimum be levied a federal income tax. The federal tax rate on earnings per life-year used in this analysis is a very conservative estimate and could increase to 20–30% for individuals falling into higher tax brackets. Absent from these estimates were dollar recoveries at the state level, which may be contributory. 

Finally, we calculate that the cost of TVUS screening could increase by more than one-quarter (26.9%—UK HealthCare rates) before the economic benefit of decreased clinical management due to increased early-stage detection coupled with increased earnings and taxpayer contributions disappears. The reader should be aware that in providing these estimates, we have not placed a price on human life, but on the services that may extend life in the setting of ovarian cancer.

## Figures and Tables

**Figure 1 diagnostics-10-00067-f001:**
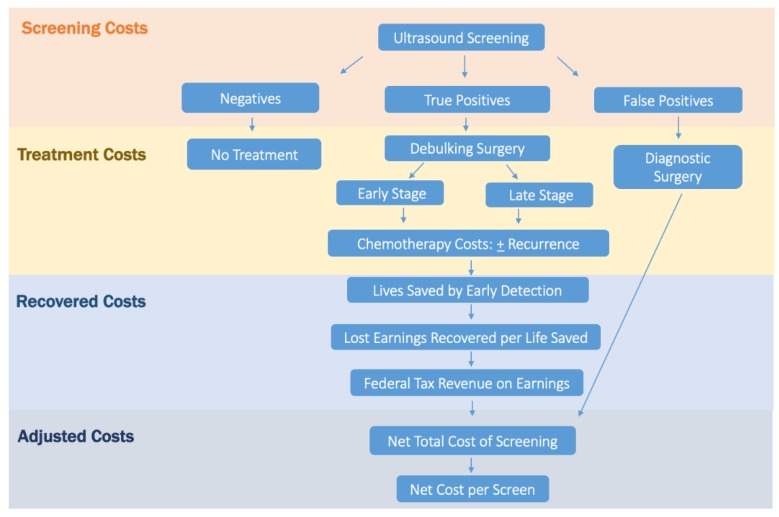
Schematic representation of the amended step-wise approach used in this analysis to estimate the adjusted net total cost and adjusted net cost per screen of the Kentucky trial.

**Figure 2 diagnostics-10-00067-f002:**
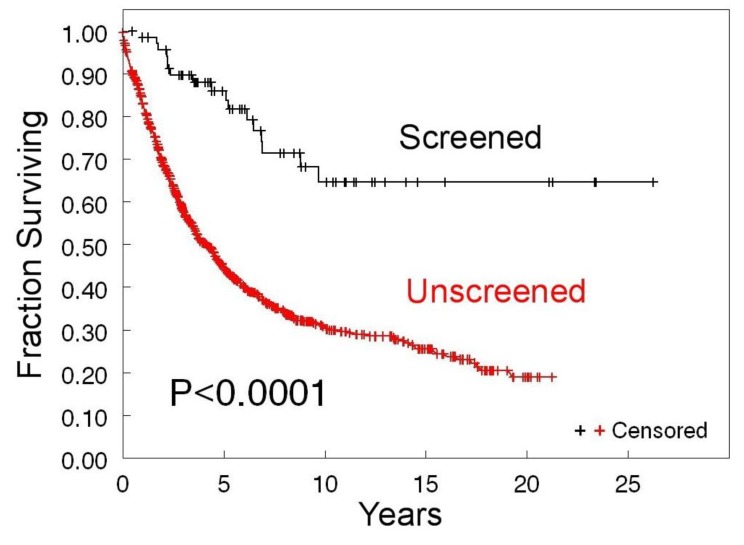
Kaplan–Meier disease-specific survival of screened and unscreened women with epithelial ovarian cancer by method. Vertical lines indicate censored points.

**Table 1 diagnostics-10-00067-t001:** Estimates of the costs associated with the Kentucky trial accounting for the direct cost of ultrasound screening and surgical costs due to false-positive cases.

Cost of Screens and False-Positive Cases	
A. Cost/screen	$56
B. No. of screens	326,998
C. Cost of screens *	$18,311,888
D. No. of true-positive results	127
E. No. of false-positive results	381
F. Positive Predictive Value (PPV)	25%
G. Surgical cost/false-positive case	$7954
H. Total surgical cost of false-positive cases **	$3,030,474
I. Total cost of screens and false-positive cases ***	$21,342,362

* row A × row B; ** row E × row G; *** row C + row H.

**Table 2 diagnostics-10-00067-t002:** Estimated cost of treating screen-detected and clinically detected ovarian cancer using the Kentucky cost estimates (A) and the national cost estimates (B).

**A. Treatment Costs: Kentucky**				
	i. Screen-Detected Stage I–II (*N* = 48)	ii. Screen-Detected Stage III–IV (*N* = 28)	iii. Clinically Detected Stage I–II (*N* = 22)	iv. Clinically Detected Stage III–IV (*N* = 54)
a. Primary Treatment Cost ^+^	$1,606,272	$936,992	$736,208	$1,807,056
	Recurrence (*N* = 7)	Recurrence (*N* = 23)	Recurrence (*N* = 3)	Recurrence (*N* = 52)
b. Recurrence Treatment Cost ^#^	$1,124,613	$3,695,157	$481,977	$8,354,268
c. Total Treatment Cost by Stage *	$2,730,885	$4,632,149	$1,218,185	$10,161,324
d. Treatment Cost All Stages **	$7,363,034		$11,379,509	
e. Treatment Cost Advantage of Screening *** $4,016,475				
**B. Treatment Costs: National (U.S.)**				
	i. Screen-Detected Stage I–II (*N* = 48)	ii. Screen-Detected Stage III–IV (*N* = 28)	iii. Clinically Detected Stage I–II (*N* = 22)	iv. Clinically Detected Stage III–IV (*N* = 54)
a. Primary Treatment Cost ^+^	$4,494,336 ($2,991,312–$3,923,920)	$2,621,696 ($1,744,932–$3,923,920)	$2,059,904 ($1,371,018–$3,083,080)	$5,056,128 ($3,365,226–$7,567,560)
	Recurrence (*N* = 7)	Recurrence (*N* = 23)	Recurrence (*N* = 3)	Recurrence (*N* = 52)
b. Recurrence Treatment Cost ^#^	$427,014 ($203,196–$927,542)	$1,403,046 ($667,664–$3,047,638)	$183,006 ($87,084–$397,518)	$3,172,104 ($1,509,456–$6,890,312)
c. Total Treatment Cost by Stage *	$4,921,350 ($3,194,508–$7,654,262)	$4,024,742 ($2,412,576–$6,971,558)	$2,242,910 ($1,458,102–$3,480,598)	$8,228,232 ($4,874,682–$14,457,872)
d. Treatment Cost All Stages **	$8,946,092 ($5,607,084–$14,625,820)		$10,471,142 ($6,332,784–$17,938,470)	
e. Treatment Cost Advantage of Screening *** $1,525,050 ($725,700–$3,312,650)				

^+^ Includes cost of cytoreductive surgery and adjuvant or neoadjuvant carboplatin/paclitaxel during the first year of treatment; ^#^ Includes expenses incurred during two additional years of treatment for each subject with recurrence; * row a + row b; ** row c column i + row c column ii or row c column iii + row c column iv; *** row d column iii and iv − row d column i and ii.

**Table 3 diagnostics-10-00067-t003:** Patient lives saved and total life-years gained due to implementation of the Kentucky screening program.

Lives Saved and Life-Years Gained Due to Screening	
A. The 10-year survival screen-detected cases (*N* = 76)	52 (68%)
B. The 10-year survival clinically detected cases (*N* = 76)	26 (34%)
C. Patient lives saved *	26
D. Mean age screen-detected women (years)	65
E. Mean age female life expectancy, United States (years)	81
F. Mean life-years gained/life saved **	16
G. Total life-years gained ***	416

* row A − row B; ** row D − row E; *** row C × row F.

**Table 4 diagnostics-10-00067-t004:** Kentucky trial screening costs per life-year gained adjusted for the surgical cost of false positives and the cost advantage of treating more early-stage disease.

Adjusted Kentucky Trial Screening Program Cost per Life-Year Gained		
	Kentucky	National (U.S.)
A. Base cost of screens/life-year gained *	$44,019	$44,019
B. Surgical cost of false-positive cases/life-year gained **	$7285	$7285
C. Cost advantage of treating more early-stage disease/life-year gained ***	$9655	$3666 ($1,744–$7,963)
D. Adjusted screening cost/life-year gained ****	$41,649	$47,638 ($43,341–$49,560)

* Table 1 row C ÷ Table 3 row G; ** Table 1 row H ÷ Table 3 row G; *** Table 2 row e ÷ Table 3 row G; **** row A + row B − row C.

**Table 5 diagnostics-10-00067-t005:** Estimate of the lost earnings gained and federal income tax dollars recovered on those earnings due to screening.

Total Earnings Gained and Recovered Federal Tax Dollars Due to Screening	
A. Lost earnings/death [17]	$136,500 ($132,518–$140,643)
B. Patient lives saved	26
C. Total earnings gained due to screening *	$3,549,000 ($3,445,468–$3,656,718)
D. Average federal income tax rate on earnings gained	0.108
E. Total federal income tax on earnings **	$383,292 ($372,111–$394,926)

* row A × row B; ** row C × row D.

**Table 6 diagnostics-10-00067-t006:** Summary of the total net cost of the Kentucky screening program from 1987 to 2019 when accounting for the cost of screens, total surgical cost of false-positive cases, cost advantage of treating more early-stage disease, lost earnings gained and federal tax dollars recovered by women saved due to screening.

Adjusted Net Total Screening Cost		
	Kentucky	National (U.S.)
A. Cost of screens (Table 1)	$18,311,888	$18,311,888
B. Total surgical cost of false-positive cases (Table 1)	$3,030,474	$3,030,474
C. Total cost advantage of treating earlier detected cases (Table 2)	$4,016,475	$1,525,050 ($725,700–$3,312,650)
D. Total earnings gained due to screening (Table 4 row C)	$3,549,000 ($3,445,468–$3,656,718)	$3,549,000 ($3,445,468–$3,656,718)
E. Total federal income tax dollars recovered on earnings gained (Table 4 row E)	$383,292 ($372,111–$394,926)	$383,292 ($372,111–$394,926)
F. Adjusted total net screening cost *	$13,393,595 ($13,508,308–$13,274,243)	$15,885,020 ($13,978,068–$16,799,083)

* (row A+ row B) − (row C + row D + row E).

**Table 7 diagnostics-10-00067-t007:** Estimate of the cost per screen during the Kentucky screening trial adjusted for surgical cost of false-positive cases, cost advantage of treating earlier stage disease and lost earnings and federal tax dollars recovered due to lives saved.

Adjusted Cost per Screen		
	Kentucky	National (U.S.)
Screens alone (Table 1 row A)	$56.00	$56.00
Screens and surgical cost of false-positive cases *	$65.27	$65.27
Screens and cost advantage of treating earlier detected cases **	$43.72	$51.34 ($45.87–$53.78)
Screens and recovered earnings and federal tax dollars ***	$43.97 ($43.61–$44.33)	$43.97 ($43.61–$44.33)
Adjusted net cost ****	$40.96 ($40.59–$41.31)	$48.58 ($42.75–$51.37)

* (Table 6A + 6B)/326,998; ** (Table 6 row A − Table 6 row C)/326,998; *** (Table 6 row A − Table 6 row D + Table 6 row E)/326,998; **** (Table 6 row F)/326,998.

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
