# Peer review of "Assessing the Costs of Screening for Ovarian Cancer in the United States: An Evolving Analysis"

_diagnostics, 2020, doi:10.3390/diagnostics10020067_

Round 1
Reviewer 1 Report
The authors provided a detailed analysis of the cost of screening for ovarian cancer in the United States. The work carried out is thorough and provides great insights in to the importance of screening programs. I appreciate the study.
Author Response
Thank you. We appreciate you taking the time to review our study. Your insights are valuable and we look forward to publication of this manuscript in Diagnostics after some minor spelling and grammar edits.
Reviewer 2 Report
In this manuscript, Gorski at al. assess the screening costs of ovarian cancer in the United States. The primary objective of this study is to provide an updated analysis of the cost of screening for ovarian cancer in the United States. The authors used updated information from the University of Kentucky Ovarian Cancer Screening Trial in conjunction with new modifying factors such as: U.S. national estimates of the cost of care, recently published estimates of earnings lost due to ovarian cancer death and estimates of federal income taxes paid on those earnings.
I commend the authors on this very well written manuscript. They describe their approach and results comprehensively, and discuss their findings very clearly. I do not have any major comments only a couple of minor suggestions:
Please double check that all abbreviations are defined before they are used (e.g., PLCO).
It may be helpful to include a diagram summarizing all factors included in this cost analysis.
Author Response
Thank you for your kind feedback. We have made the appropriate edits based on your suggestions and feel that the manuscript is now clearer and has benefited tremendously from your insights.
Please double check that all abbreviations are defined before they are used (e.g., PLCO).
We have adjusted the second paragraph of the introduction to now read:
Over the last three decades, a concerted effort has been made to develop an accurate, reliable and cost effective screening modality to detect early stage ovarian cancer. Four major trials have evaluated the utility of using transvaginal ultrasound to detect ovarian cancer including the Kentucky trial [4], the Prostate, Lung, Colorectal and Ovarian (PLCO) Cancer Screening Randomized Controlled Trial [5], the UK Collaborative Trial of Ovarian Cancer Screening (UKCTOCS) [6] and the Shizuoka Cohort Study of Ovarian Cancer Screening (SCSOCS) [7] trial. Three of these trials (Kentucky, UKCTOCS and SCSOCS) report a shift to early stage detection, while both the Kentucky and UKCTOCS report a survival advantage for incident cases due to screening [8,9].
We have also looked through the manuscript to ensure that all abbreviations are defined as suggested.
It may be helpful to include a diagram summarizing all factors included in this cost analysis.
We have built a schematic representation of our methodology and have included it at the end of the Materials and Methods sub-section "2.2 Cost Analysis" as the new "Figure 1". Please see attached for the diagram and caption.
Thank you for your suggestions. They have vastly improved this manuscript.
